Age, growth and natural mortality of coney (Cephalopholis fulva) from the southeastern United States

Burton Michael L. michael.burton@noaa.gov
Potts Jennifer C.
Carr Daniel R.
Beaufort Laboratory, Southeast Fisheries Science Center, National Marine Fisheries Service, NOAA , Beaufort, NC , United States
Esteban María Ángeles
Electronic publication date: 2015 Mar 19
Publication date: 2015
Volume: 3
Electronic Location ID: e825
Received 2014 Dec 16; Accepted 2015 Feb 17
Copyright year: 2015
License: This is an open access article, free of all copyright, made available under the Creative Commons Public Domain Dedication. This work may be freely reproduced, distributed, transmitted, modified, built upon, or otherwise used by anyone for any lawful purpose.
License URL: https://creativecommons.org/publicdomain/zero/1.0/

Keywords: Serranidae, Life history, Age and growth, Natural mortality

Funding: National Marine Fisheries Service, Southeast Fisheries Science Center, Miami, FL This work was funding by the National Marine Fisheries Service, Southeast Fisheries Science Center, Miami, FL. The Southeast Fisheries Science Center had no role in the design or execution of the study but did have the final say in the decision to publish the manuscript (did it meet strict scientific guidelines and quality).

==============================
Coney (Cephalopholis fulva) sampled from recreational and commercial vessels along the southeastern coast of the United States in 1998–2013 (n = 353) were aged by counting opaque bands on sectioned sagittal otoliths. Analysis of otolith edge type (opaque or translucent) revealed that annuli formed in January–June with a peak in April. Coney were aged up to 19 years, and the largest fish measured 430 mm in total length (TL). The weight-length relationship was ln(W) = 3.03 × ln(TL) − 18.05 (n = 487; coefficient of determination [r2] = 0.91), where W = whole weight in kilograms and and TL = total length in millimeters. Mean observed sizes at ages 1, 3, 5, 10, and 19 years were 225, 273, 307, 338, and 400 mm TL, respectively. The von Bertalanffy growth equation for coney was Lt = 377 (1 − e(−0.20(t+3.53))). Natural mortality (M) estimated by Hewitt and Hoenig’s longevity-based method which integrates all ages was 0.22. Age-specific M values, estimated with the method of Charnov and others, were 0.40, 0.30, 0.26, 0.22, and 0.20 for ages 1, 3, 5, 10, and 19, respectively.

Introduction

The coney (Cephalopholis fulva Linnaeus 1758) is a small to medium-sized member of the grouper family (Serranidae) widely distributed throughout the tropical and subtropical waters of the western Atlantic from North Carolina and Bermuda through southern Brazil, including the Gulf of Mexico and Caribbean Sea (Heemstra & Randall, 1993). Coney inhabit shallow to moderately deep coral reefs (Nagelkerken, 1981) and associated rocky ledge habitats.

Coney are of minor importance to the commercial and recreational fishing sectors in southeastern United States (SEUS) Atlantic Ocean waters. Within the recreational sector, annual landings of coney are minimal compared to other serranids. Estimated landings of coney from headboats (vessels engaged in recreational fishing, usually carrying from 16–100 anglers) sampled by the Southeast Region Headboat Survey (SRHS), which is administered by the Beaufort Laboratory of the Southeast Fisheries Science Center (SEFSC), National Marine Fisheries Service (NMFS), averaged 205 fish totaling 114 kg annually from 1981–2012 for SEUS waters (K Brennan, 2014, unpublished data). Estimated landings from private recreational boats and charter boats, the other component of the recreational sector, averaged 1,473 fish annually during 1982–2011 in the SEUS. However, landings by this sector in Puerto Rico were substantially more important, averaging 32,768 fish annually from 2000–2012 (T Sminkey, 2014, unpublished data). Commercial sampling programs in the SEUS accumulate small groupers such as coney in an unclassified grouper category, and thus commercial landings statistics of coney are unavailable (D Gloeckner, SEFSC Miami, FL., pers. comm., 2014). The majority of SEUS recreational landings of coney occur in Florida with the Carolinas contributing an average of only 58 fish annually from the headboat and private recreational sectors combined.

Coney are currently managed by the South Atlantic Fishery Management Council (SAFMC) through inclusion in a shallow water grouper category with a closed season from January–April and inclusion in a daily aggregate bag limit of three groupers per person for recreational fishermen (SAFMC , 2014).

Published studies on the aspects of life history of coney from the western Atlantic are limited. Potts & Manooch (1999) reported on the age and growth of the species from headboat samples in SEUS waters. Trott (2006) and Trott & Luckhurst (2007) examined aspects of the biology and population status of the species in Bermuda waters. Araujo & Martins (2006) and Araujo & Martins (2009) studied the age and growth and population dynamics of coney from the central coast of Brazil.

We revisited coney from the SEUS because it is one of the 60 species managed by the SAFMC and little new biological information on the species has been published in recent years. This study provides information on life history parameters for coney collected from the commercial and recreational fisheries of the SEUS and compares these new parameter estimates to previous life history studies.

Materials and methods

Age determination

Coney were sampled from fisheries landings along the SEUS coast (North Carolina through the Florida Keys) from 1998 to 2013 by port agents employed by the SRHS, which samples headboats, and the SEFSC Trip Interview Program (TIP), which samples commercial fisheries landings. Both surveys operate using a random sample design methodology. All specimens used in this study were killed as part of legal fishing operations and were already dead when sampled by the port agents, thus all research was conducted in accordance with the Animal Welfare Act (AWA) and with the U.S. Government Principles for the Utilization and Care of Vertebrate Animals Used in Testing, Research, and Training (USGP) OSTP CFR May 20, 1985, Vol. 50, No. 97. The majority of specimens from both fisheries were captured by conventional, vertical hook-and-line gear. Total length (TL) of specimens was recorded in millimeters. Whole weight (W, kg) was recorded for fish landed in the headboat fishery. Whole weights were unavailable for fish landed commercially because they were eviscerated at sea. Because of predetermined sampling protocols and time constraints imposed by their workload, as well as the protogynous nature of coney and their small size, samplers were not required to determine the sex of sampled coney due to uncertainty in macroscopic staging.

Sagittal otoliths were removed from 367 coney and stored dry in coin envelopes. Otoliths were sectioned in the transverse dorso-ventral plane on a low-speed saw, following the methods of Potts & Manooch (1995). The sections were mounted on microscope slides with thermal cement and covered with mounting medium before analysis. The sections were viewed under a dissecting microscope at 12.5 × with transmitted light. Readings were taken from the dorsal lobe of the otolith. Initial ring counts were made along the sulcal groove (vertically from the core) and then confirmed by following the rings out to the lateral plane (horizontal from the core). Each sample was assigned a ring count equal to the number of opaque zones. Two readers interpreted otolith sections. To ensure consistency between readers in the interpretation of growth structures, each individual read all 367 slides, then we calculated between-reader indices of average percent error (APE) following the methodology of Campana (2001).

Edge analysis was used to validate the annual deposition of the opaque zone in coney otoliths. The edge type of the otolith section was noted: 1, opaque zone forming on the edge of the otolith section; 2, narrow translucent zone on the edge, generally <30% of the width of the previous translucent zone; 3, moderate translucent zone on the edge, generally 30–60% of the width of the previous translucent zone; 4, wide translucent zone on the edge, generally >60% of the width of the previous translucent zone (Harris et al., 2007). Frequency of all edge types by month was then plotted to determine the period of opaque zone deposition. Edge analysis is based on the assumption that there is a yearly sinusoidal cycle in the plot of relative frequency of edge types over time (Campana, 2001). On the basis of this frequency analysis of edge type, all samples were assigned a chronological, or calendar, age obtained by increasing the opaque zone count by one if the fish was caught before that increment was formed and had an edge with a translucent zone that was moderate to wide (type 3 or 4). All fish caught after opaque zone formation would have had a chronological age equivalent to the opaque zone count.

Growth

Von Bertalanffy (1938) growth parameters were estimated from the observed length-at-age data, using the chronological age. Parameters were derived using PROC NLIN, a non-linear regression procedure using least squares estimation and the Marquardt iterative algorithm option, in SAS statistical software (vers. 9.3; SAS Institute , 1987). Appropriate statistical tests were employed to examine differences in mean size and age by fishery sector (Student’s t-test [P < 0.05] for normally distributed data; non-parametric Kolmogorov–Smirnov test [P < 0.05] for skewed distributions). We also examined mean length-at-age data by sector (recreational versus commercial) to determine if pooling of data was appropriate (i.e., there were no significant differences in length- at-age by sector).

Body–size relationships

We regressed fish whole weight (kg) on fish TL (mm) using data for all coney measured by the SRHS from 1976–2014 (n = 487). We evaluated a nonlinear fit, using PROC NLIN in SAS (SAS Institute , 1987), and a linearized fit of the log-transformed data, examining the residuals to determine which regression was appropriate.

Natural mortality

We estimated the instantaneous rate of natural mortality (M) using 2 methods.

(1) Hewitt & Hoenig’s (2005) longevity mortality relationship: M≈4.22/tmax,

where tmax is the maximum age of the fish in the sample.

(2) Charnov, Gislason & Pope (2013) method, which uses the von Bertalanffy growth parameters: M=L/L∞−1.5×K,

where L∞ and K are the von Bertalanffy growth equation parameters (asymptotic length and growth coefficient) and L is fish length at age. The Hewitt & Hoenig (2005) method uses life span or longevity to generate a single point estimate, and it is an improvement to the original equation of Hoenig (1983). The newer Charnov method, which incorporates growth parameters, is an improvement to the empirical equation of Gislason et al. (2010) and is based on evidence that M decreases as a power function of body size. The Charnov method generates age-specific rates of M and is currently in use in Southeast Data Assessment and Review (SEDAR) stock assessments (E Williams, pers. comm., 2013).

Results

Age determination

A total of 367 sagittal otoliths of coney were sectioned. The distribution of samples by state and fishery sector is shown in Table 1. The majority of samples (75%) came from the commercial sector in the Carolinas. Only 9% of aging samples were from Florida, with the majority of those samples from headboats. Opaque zones were counted on 353 (96.2%) of the 367 sectioned otoliths. Sections from the other 14 otoliths (3.8%) were judged illegible and were excluded from this study.

Table 1 Number of samples of sagittal otoliths that were used for age and growth study of coney (Cephalopholis fulva) collected from 1998–2013 primarily from fisheries landings along the coast of the southeastern United States.

Samples were collected in the following states: North Carolina (NC), South Carolina (SC), and Florida (FL).

State	Commercial	Recreational	
NC	242	0	
SC	32	2	
FL	7	74	

For our analysis of increment periodicity, we assigned an edge type to all 353 samples. Opaque zones on the otolith marginal edge occurred in samples collected from January to June (Fig. 1); occurrence of opaque zones was highest from March to June with peak formation in April (Fig. 1). We concluded that opaque zones on coney otoliths were annuli. Chronological ages resulting from edge analysis were assigned as follows: for fish that were caught from January to June and had an edge type of 3 or 4, the chronological age was the annuli count plus one; for fish that were caught during the same period and had an edge type of 1 or 2 and for fish that were caught from July to December, the chronological age was equivalent to the annuli count.

Figure 1 Monthly percentages of all edge types for coney (Cephalopholis fulva) collected from the southeastern United States in 1998–2013.

Edge codes: 1, opaque zone on edge, indicating annulus formation; 2, small translucent zone, <30% of previous increment; 3, moderate translucent, 30–60% of previous increment; 4, wide translucent, >60% of previous increment.

Growth increments of coney were moderately easy to interpret. Based on Campana’s (2001) acceptable value of APE (5%), agreement was good between the two readers (MLB–JCP APE = 6.4%, n = 353). Percent agreement values between the two readers were moderate (49%) but increased for estimates within (±) 1 year (87%) and (±) 2 years (92%). These results indicate acceptable between-reader agreement.

Growth

Coney in this study (n = 353) ranged from 217 to 430 mm TL and from age 1 to 19. There were only 11 (3%) fish older than age 12 (Table 2). Length and age distributions by sector are shown in Fig. 2. Visual examination of these size and age frequency distributions identified apparent differences by sector. Modal lengths were 325 mm TL for the commercial sector and 275 mm TL for the recreational sector (Fig. 2A). Mean lengths by sector were significantly different: 325 mm TL (±standard error [SE] 2.1) for the commercial sector versus 280 mm TL (SE 3.6) for the recreational fishery (Student’s t test: t = 11.00; P = 0.0001) (Fig. 2A). The modal age frequencies were age-5 years and age-3 years for the commercial and recreational sectors, respectively (Fig. 2B). Due to the visually skewed appearance of age frequency distributions, we tested for normality using PROC UNIVARIATE (SAS Institute , 1987) and found age frequency data to be non-normally distributed (Wilcoxon signed rank test, P < 0.0001). Mean ages were significantly different between fisheries: 6.6 years (SE 0.17) versus 4.4 years (SE 0.32) for commercial and recreational sectors, respectively (non-parametric Kolmogorov–Smirnov test; D = 0.51, P < 0.0001). Visual examination of mean size-at-age of coney by fishery sector (Fig. 3) revealed no significant difference in growth by sector for ages 2–11, as indicated by overlapping error bars. Additionally, we performed an analysis of covariance (ANCOVA) of length-at-age by sector, using age as the covariate, and found no significant differences in size at age between sectors for six out of nine ages for which we had adequate sample size for comparison. On the basis of these results, we pooled data across sectors. The resulting von Bertalanffy growth equation was: Lt=3771−e−0.20t+3.53

for all fishery sectors combined (n = 353) (Table 3 and Fig. 4).

Figure 2 Distributions of (A) length frequency and (B) age frequency, by fishing sector, for aging samples of coney (Cephalopholis fulva) collected from the southeastern United States in 1998–2013.

Figure 3 Comparison of mean size at age, measured in total lengths (TLs), for coney (Cephalopholis fulva), by fishery sector, sampled from the southeastern United States in 1998–2013.

Figure 4 Observed and predicted lengths at age, measured in fork lengths (FLs), for coney (Cephalopholis fulva) sampled from the southeastern United States in 1998–2013.

Lt, length at age t; t0, time when length is zero.

Table 2 Observed and predicted mean total length (TL), measured in millimeters, and natural mortality at age (M) data for coney (Cephalopholis fulva) collected in 1998–2013 along the coast of the southeastern United States.

Standard errors of the means (SE) are provided in parentheses.

Age	n	Mean TL (±SE)	TL range	Predicted TL	M	
1	1	225	–	225	0.49	
2	11	255 (6)	218–287	252	0.40	
3	34	273 (4)	237–325	275	0.34	
4	59	294 (4)	220–362	293	0.31	
5	66	308 (3)	227–377	308	0.28	
6	56	321 (4)	250–392	321	0.26	
7	35	329 (6)	217–380	331	0.25	
8	37	342 (4)	287–395	339	0.24	
9	14	357 (6)	310–402	346	0.23	
10	9	338 (9)	310–390	352	0.22	
11	8	360 (13)	330–430	356	0.22	
12	9	353 (10)	281–380	360	0.21	
13	1	345	–	363	0.21	
14	1	370	–	366	0.21	
15	1	285	–	368	0.21	
16	2	356	347-365	369	0.21	
17	3	388 (1)	385–390	371	0.21	
18	2	393 (11)	382–404	372	0.20	
19	1	400	–	373	0.20	

Table 3 Comparison of life history parameters of coney (Cephalopholis fulva) from various studies.

	Parameter	
Study	L ∞	K	t 0	Peak opaque edge	n	Maximum age	
Potts & Manooch (1999)—SEUS	372	0.32	−0.20	March	55	11	
Trott (2006)—Bermuda	281	0.20	−1.21	–	997	28	
Araujo & Martins (2006)—Brazil	316	0.14	−5.74	April–June	705	25	
Burton et al. (current study)	377	0.20	−3.53	April–May	353	19	
Notes.

L∞ asymptotic length

K growth coefficient

t0 theoretical age at length of zero

n sample size

SEUS southeastern United States

TL total length

Body–size relationships

Statistical analyses revealed a multiplicative error term (variance increasing with size) in the residuals of the W–TL relationship, indicating that a direct nonlinear fit was not appropriate. We used a linearized ln-transform fit of the data, resulting in the following regression to describe the relationship: lnW=3.03×lnTL−11.15n=487,r2=0.91

where r2 is the coefficient of determination. This equation was transformed back to the form W=a×TLb

after adjustment of the intercept for log-transformation bias with the addition of one-half of the mean square error (MSE) (Beauchamp & Olson, 1973), resulting in this relationship:

W = 1.457 × 10−8TL3.03(n = 487; MSE = 0.022) (Fig. 5).

Figure 5 Scatter plot of weight–length relationship for coney (Cepohalophilis fulva) sampled from the southeastern USA (W, weight; MSE. mean square error)

Natural mortality

The method of Hewitt & Hoenig (2005), which uses maximum age or life span (age-19 years in this study), estimated that M was 0.22. The method of Charnov, Gislason & Pope (2013), which produces age-specific estimates of M using von Bertalanffy growth parameters, estimated M of 0.49 for age-1, 0.28 for age-5, 0.22 for age-10, and 0.20 for age-19 (Table 2).

Discussion

Coney were relatively easy to age and we found consistent agreement between readers with low APE values. Because our study included samples from the commercial and recreational sectors of the SEUS fishery, we believe that the results are likely to be more robust and representative of the SEUS coney population than those presented by Potts & Manooch (1999), due to our larger sample size and wider fishery sector coverage. The results of this study represent the best contemporary information on the longevity, growth and natural mortality of coney from SEUS waters.

The otolith edge analysis that we conducted strongly indicated that coney deposit one annulus per year from January to June with peak annulus formation in April. This result compares favorably to findings in other studies showing that peak annulus formation occurred in March in coney caught by the headboat fishery of the SEUS (Potts & Manooch, 1999) and in April through June for fish from the central coast of Brazil (Araujo & Martins, 2006) (Table 3).

Body–size relationships were nearly identical for coney from this study (W = 1.46 × 10−8TL3.03), the previous SEUS study by Potts & Manooch (1999) (W = 2.59 × 10−8TL2.94), and the study by Araujo & Martins (2006) from Brazil (W = 2.0 × 10−8TL2.97).

The growth rate for coney was relatively slow compared to other groupers, with coney attaining an average observed size of 294 mm TL by age 4 (Table 2). Similarly, Potts & Manooch (1999) observed a mean size of 278 mm TL by age 4. Growth of fish in our study slowed after age 4; coney reached 342 mm TL by age 8, then averaged annually only increments of 5 mm through age 19.

Our theoretical growth curve fit the observed data well (Fig. 4). Growth parameters estimated in our study compare most closely with the previous SEUS study (Potts & Manooch, 1999; Fig. 6) for fish age 6 and older. The lack of small coney in our samples (smallest fish was 218 mm TL versus 150 mm TL in Potts & Manooch (1999)) explains differences in the early years of growth among studies (Fig. 6). Greater numbers of smaller fish no doubt help define the growth curve for the younger ages. Also, Potts & Manooch (1999) used back-calculated lengths to model growth; this approach allowed the model to more adequately capture initial growth of coney. Though the size range of the coney in our study overlapped with the mean lengths at age of those samples available in the Potts & Manooch (1999) study, the fish in our sample were larger at age for ages 2–5. The theoretical maximum length of coney in our study (L∞ = 377 mm TL) was similar to that reported by Potts & Manooch (1999; 372 mm TL) but much larger than the L∞ estimated for Bermuda (Trott, 2006, 281 mm TL) and Brazil (Araujo & Martins, 2006, 316 mm TL). On the other hand, the studies from Bermuda (Trott, 2006, maximum age 27) and Brazil (Araujo & Martins, 2006, maximum age 25) included fish that were much older than our eldest fish (age 19). Perhaps fishing pressure in the SEUS is greater than in these other areas, contributing to a truncated age structure in the SEUS. The lack of older fish in the previous SEUS study (Potts & Manooch, 1999) is likely attributable to the lack of any commercially landed samples in the study.

Figure 6 Comparison of von Bertalanffy growth curves for coney (Cephalopholis fulva) from various studies in western Atlantic waters.

Natural mortality of wild populations of fishes is difficult to measure. A single estimate of M for the entire life span of a fish is unreasonable, except for fish that have attained a size that renders them invulnerable to high predation rates. The Hewitt & Hoenig (2005) estimate of M is based on the maximum age attainable in an unfished population. In this sense, the point estimate of M, derived using the method of Hewitt & Hoenig (2005), can serve as a lower boundary on the estimate of M derived for older ages by an age-varying method. The maximum observed age for coney in our study was age 19, in the middle of the range of maximum ages estimated by others (Table 3). Our estimates seem reasonable given that our age-specific estimate of M (0.20) for the older ages that was derived using Charnov, Gislason & Pope (2013) compares closely with the point estimate of M(0.22) found using the method of Hewitt & Hoenig (2005) (Table 2).

This study of coney in the SEUS has confirmed the findings of previous studies that otolith sections of coney are reliable structures for aging. Moreover, growth rings on coney sagittae are laid down once a year in spring and growth is generally slow throughout life, as evidenced by the low value of K, the von Bertalanffy growth coefficient. Our estimates of M are reasonable for a fish with a moderate life span and longevity of age 19. We believe the results of this study accurately describe the fished population of coney in the offshore waters of the SEUS. While the overall landings of this species in the commercial and recreational fisheries of the SEUS make it an unlikely candidate for a stock assessment by NMFS, our assembled data would be valuable inputs into multispecies- or ecosystem-based modeling efforts, either as stand-alone species data or in defining more inclusive functional groups of species (Christensen et al., 2009). A more likely use of these data would be applying it to studies of the population dynamics of U. S. Caribbean stocks (U. S. Virgin Islands and Puerto Rico). The U. S. Caribbean is typically a data poor region and studies from the SEUS could be used as proxies in analyses for the region. However, precautionary management should dictate that for the purposes of managing local stocks, local studies of growth, reproduction, etc., are used. The difference in growth parameters between various regions found in this study highlights the problems inherent with drawing scientific conclusions using data from outside the region.

Supplemental Information

Supplemental Information 1 Raw Data file for Coney age growth study

Click here for additional data file.

We gratefully acknowledge the many headboat and commercial port agents who obtained samples over the years. Their efforts made this study possible. J Smith, R Cheshire, and two anonymous reviewers provided valuable reviews that greatly improved this manuscript.

Additional Information and Declarations

Competing Interests

Author Contributions

Animal Ethics

The authors declare there are no competing interests.

Michael L. Burton conceived and designed the experiments, performed the experiments, analyzed the data, wrote the paper, prepared figures and/or tables, reviewed drafts of the paper.

Jennifer C. Potts performed the experiments, analyzed the data, contributed reagents/materials/analysis tools, wrote the paper, prepared figures and/or tables, reviewed drafts of the paper.

Daniel R. Carr analyzed the data, contributed reagents/materials/analysis tools.

The following information was supplied relating to ethical approvals (i.e., approving body and any reference numbers):

All research was conducted in accordance with the Animal Welfare Act (AWA) and with the U.S. Government Principles for the Utilization and Care of Vertebrate Animals Used in Testing, Research, and Training (USGP) OSTP CFR May 20, 1985, Vol. 50, No. 97. The study was conducted on cold-blooded vertebrates (fish) which were already dead when collected and processed by the samplers for this study.

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
