# Peer review of "Age, growth and natural mortality of coney (Cephalopholis fulva) from the southeastern United States"

_PeerJ, doi:10.7717/peerj.825_

## Round 0.1 · original submission · Major Revisions

All the suggestions made by the reviewers have to be considered in the new version of the mansucript.

Reviewer 1 ·

Basic reporting

This study contains three main topics: 1) age determination using otolith opaque/translucent rings, 2) fitting growth function, and 3) estimating mortality rate for coney fish. These provide basic information on life history parameters which would be useful for accumulation of ecological aspects of this species. However, I feel that the information presented in it is far more appropriate for a specialist journal dealing with fish biology and ichthyology. Your discussion further supports my argument because the results are only relevant for coney fish in particular region and very minor importance for fisheries management.

Experimental design

Otolith preparation, counting annuli, fitting statistical models were straightforward and appropriately described. However, in some part, explanation for sampling and analysis are too short and insufficient to judge the validity of the study. For instance, otolith was categorized into four stages, but lacks sample picture of each category. As map of the sampling locations and areas were not shown, I cannot evaluate whether the present data represent entire coney population in SEUS area. Further, analyses was partly based on very subjective method (e.g. visual test for growth curve difference). Thereby it seems as if the authors lack detailed knowledge of otolith growth dynamics and relating experimental technics.

Validity of the findings

The most important findings is in the periodicity of otolith opaque and translucent zones. The authors concluded that coney otolith deposit one annulus per year. I am not conceived the idea, because only small numbers of fish (approximately 30% of entire samples) have Edge 1 stage otolith during January-June. Rather, the result indicates most of coney fish does not deposit opaque zone during these months, and therefore otolith annulus formation could occur once every three years. The authors need to pay attention to this discrepancy. As all following analysis are based on the assumption of otolith annulus periodicity, I could not recognize the validity of entire conclusions.

Additional comments

L15-49 The introduction is short and does not provide much background on these species (nor comparison with other species). I think some context here is needed to better understand these results.
L67 Which side of otolith was used for?
L67 Vertical or horizontal? Please specify.
L76 Sample pictures showing each stage should be provided.
L88 ‘PROC NLIN’ seems not general method. More general statistical procedure, such as the least square, maximum likelihood, or Bayesian framework, should be used. If ‘PROC NLIN’ is more appropriate for the study, please provide the reason.
L89 The t-test compares the mean of normally distributed samples, but age frequency is positively skewed.
L126 Only small numbers of otolith with opaque zone were collected. This may denied periodicity of opaque and translucent zone. Or considerable numbers of fish are immigrated from other population into study site.
The authors are advised to include sentences in the discussion on the relevance of their findings for fisheries management and biology of other related species, e.g do these results alter or confirm current hypotheses and input data for fisheries assessment and management?

·

Basic reporting

No comments

Experimental design

No comments

Validity of the findings

No comments

Additional comments

This paper presents valuable age and growth information on a data poor species. I thought overall the paper was fairly well written and organized. While this species may never warrant a stock assessment, life-history studies on any species are incredibly valuable to conserving and understanding fisheries, as well as aiding management decisions. Given the nature of the study, the method of sampling and previous studies on Coney, the method of aging verification seems appropriate and warranted.

There are a few things that need clarification. In the introduction (line 24) you mention the headboats sampled by the Southeast Region Headboat Survey – can you clarify if this is considered commercial or recreational? It seems like this might be commercial from your methods (line 55) but then you go on to say that commercial landings are unavailable (line 31-33). Also, in the introduction it states (lines 33-35) that “the majority of landings occur in Florida with the Carolinas contributing an average of only 58 fish” and in the results it states that only “9% of the samples were from Florida and 75% were from the Carolinas.” If your samples are both recreational and commercial, you may want to clarify why the majority of your samples are from the Carolinas and why so few are from Florida. In the introduction (Line 47-48) it states that this study collected from the “commercial and recreational” fisheries,” but the materials and methods section (line 55) only mentions sampling from commercial fisheries. Line 60 mentions “specimens from both fisheries” but it’s never mentioned how recreational samples were collected. It seems like the SRHS might collect from recreational fisheries but this is never stated. It does state in the introduction that charter boats are considered recreational, but does not mention headboats. There are a lot of very similar acronyms (starting with S, about the same length). I think it would help the paper if you used southeastern United States Atlantic Ocean water and then used referred to it as the Atlantic, opposed to adding another acronym. Lastly, there were a few instances where Coney was not capitalized.

Minor edits:
Line 105: extra space after Charnov
Line 187: Missing a closing parentheses after (w=2.0….
Line 247: The journal is in a different format then the rest of the references
Line 254: Extra period after Biology
Line 263: Extra comma after Gislason
Line 271: Date should be bold to match the rest of the references
Line 278: Extra space after date and period
Line 281: Extra periods after Caribbean and Fisheries

This paper shows important findings for Coney life history and I would accept it with minor revisions, that would help clarify some minor details.

---

## Round 0.2 · accepted · Accept

The new version of your paper has been improved after the author revisions.